# Fault Identification in Electric Servo Actuators of Robot Manipulators Described by Nonstationary Nonlinear Dynamic Models Using Sliding Mode Observers

**DOI:** 10.3390/s22010317

**Published:** 2022-01-01

**Authors:** Alexander Zuev, Alexey N. Zhirabok, Vladimir Filaretov, Alexander Protsenko

**Affiliations:** 1Laboratory of Intelligent Information Systems for Marine Robots, Institute of Marine Technology Problems, 690091 Vladivostok, Russia; zhirabok@mail.ru (A.N.Z.); ld222@inbox.ru (A.P.); 2Department of Automation and Robotics, Far Eastern Federal University, 690091 Vladivostok, Russia; 3Robotics Laboratory, Institute of Automation and Control Processes, 690041 Vladivostok, Russia; filaret@iacp.dvo.ru; 4Department of Informatics and Control in Technical Systems, Sevastopol State University, 299053 Sevastopol, Russia

**Keywords:** nonstationary nonlinear systems, faults, identification, sliding mode observers, reduced-order models

## Abstract

The problem of fault identification in electric servo actuators of robot manipulators described by nonstationary nonlinear dynamic models under disturbances is considered. To solve the problem, sliding mode observers are used. The suggested approach is based on the reduced order model of the original system having different sensitivity to faults and disturbances. This model is realized in canonical form that enables relaxing the limitation imposed on the original system. Theoretical results are illustrated by practical example.

## 1. Introduction

Different industrial equipment, in particular, robot manipulators and consumer devices often have crucial applications in their everyday life in industrial plants. Different faults can occur in this equipment caused by specific environmental conditions and by the internal plant conditions. Due to faults, the behavior of the plant components can differ considerably from the prescribed behavior. Faults can produce an unexpected change in the system dynamics or parameters, or the occurrence of unknown signals in the plant. In robots, faults can occur in different components of the system, in particular, in actuators and sensors due to the presence of electrical devices and connections [1]. To prevent critical injuries in the plant, methods of fault detection and identification should be used.

In this paper, a fault identification scheme to deal with actuator faults in robot manipulators described by nonstationary nonlinear dynamic models is considered. There are many methods of identification, one is based on sliding mode observers (SMO) and uses peculiarities of sliding motion developed in [2] and used in [3,4,5,6].

Sliding mode observers are used for unknown input estimation and fault identification (reconstruction) in different systems [7,8,9,10,11,12,13] and for fault tolerant control [14,15]. To ensure the existence of sliding motion, the system should be minimum phase; that is, the invariant zeroes of the system must be stable and the matching condition must be satisfied [16].

To relax the matching condition, two methods have been developed. In [9,17,18,19,20], a high-order sliding mode differentiator was used forming a system, which satisfied the matching condition. In [11], multiple SMOs in cascade were used based on fictitious systems. Both methods enabled solving the problem at the cost of the complicated structure of the fault identification scheme. In addition, the system should be minimum phase and large errors can occur.

The matching condition was relaxed in [21], but the estimation of fault was corrupted by the fault derivative. In [22], asymptotic convergence was not achieved, since the estimation errors were only bounded. One paper [23] relaxed the minimum phase condition and sufficient and necessary conditions, which were less restrictive than strong detectability were received. In [24], the problem of the partial unknown input reconstruction was solved under some sufficient and necessary conditions. In [25,26], the minimum phase condition was relaxed only to requiring detectability.

Nonstationary linear systems and linear parameter-varying systems were considered in [27,28,29,30,31,32]. In [27], the varying parameters were assumed to be available and perfectly measurable. In [28,29,32], the designed SMOs had parameter-varying dynamics; therefore, complicated analysis was used to proof a convergence. One paper [30] assumed that the nonstationary parameters were changed according to some known dynamical models. In [31], virtual sensors were used to solve the problem of fault tolerant control.

In our paper, to design SMO for robot manipulators described by nonstationarity dynamic models with nonsmooth nonlinearities, we do not use matching, minimum phase, and detectability conditions. In addition, to take into account non-stationarity, our procedure is based on a canonical form with constant parameters that enables simplification of the proof of the observer convergence.

It is known that significant interactions between the individual degrees of freedom (DOF) appear in the multilink manipulators. Such interactions can cause essential changes of the servo actuators parameters [33,34]. If a manipulator is free from faults, these interactions are presented in the form of forces which act on the corresponding DOF and are described by the expression
(1)$$P_{i}=H_{i}\left(q\right){\ddot{q}}_{i}+h_{i}\left(q,\dot{q}\right){\dot{q}}_{i}+M_{ei}\left(q,\dot{q},\ddot{q}\right),$$
where $P_{i}$ is the generalized force (driving torque) acting on the *i*-th DOF; $q_{i}$ is the *i*-th component of the vector $q\in R^{n}$ of the manipulator generalized coordinates; $H_{i}\left(q\right)$ is a component characterizing the inertial properties of the corresponding DOF; $h_{i}\left(q,\dot{q}\right)$ is the component of Coriolis and velocity forces; $M_{ei}\left(q,\dot{q},\ddot{q}\right)$ takes into account the gravitational forces and the external perturbation, it does not depend on the coordinate $q_{i}$ and its derivatives; i=1,…,n, *n* is the number of manipulator degrees of freedom; and $\dot{q}$ and $\ddot{q}$ are the vectors of velocity and acceleration of generalized coordinate.

We assume that all DOF of the manipulator are equipped with similar electric servo actuators with continuous current motors of independent excitation or excitation from constant magnets. In order to use the information about the fault, the servo actuator dynamics of such a DOF can be described by the following nonlinear equations with the state variables $x_{1}\left(t\right)=q_{i}\left(t\right)$, $x_{2}\left(t\right)=\omega \left(t\right)$, and $x_{3}\left(t\right)=I\left(t\right)$: (2)$$\begin{array}{ccc} {\dot{x}}_{1}\left(t\right) & = & \frac{1}{i_{r}}x_{2}\left(t\right),\\ {\dot{x}}_{2}\left(t\right) & = & -\frac{K_{v}+h^{*}\left(t\right)}{J_{H}+H^{*}\left(t\right)}x_{2}\left(t\right)+\frac{K_{m}}{J_{H}+H^{*}\left(t\right)}x_{3}\left(t\right)-\frac{M_{e}^{*}}{J_{H}+H^{*}\left(t\right)}-\frac{M_{f}}{J_{H}+H^{*}\left(t\right)}sign\left(x_{2}\left(t\right)\right)+d\left(t\right),\\ {\dot{x}}_{3}\left(t\right) & = & -\frac{K_{\omega }}{L_{m}}x_{2}\left(t\right)-\frac{R_{m}}{L_{m}}x_{3}\left(t\right)+\frac{K_{u}}{L_{m}}u\left(t\right), \end{array}$$
where $q_{i}\left(t\right)$ is the output rotation angle at the reducer output shaft; ω(t) is the output rotation velocity at the motor output shaft; I(t) is the current through the servoactuator windings; $i_{r}$ is the reducing ratio of the reducer; $J_{H}$ is the torque of inertia of the electric servoactuator rotor and of the rotating parts of the reducer; $K_{v}$ is the viscous friction coefficient, $K_{\omega }$ and $K_{m}$ are the respective coefficients of the back EMF and of the torque; $M_{f}$ is the torque of the Coulomb friction at the motor output shaft; $R_{m}$ and $L_{m}$ are the active and inductive resistances of the electric servoactuator windings, respectively; $K_{u}$ is the amplification coefficients of power amplifiers; $H^{*}\left(t\right)=H_{i}\left(t\right)/i_{r}^{2}$, $h^{*}\left(t\right)=h_{i}\left(t\right)/i_{r}^{2}$, and $M_{e}^{*}\left(t\right)=M_{ei}\left(t\right)/i_{r}$ are, respectively, the values of $H_{i}\left(t\right)$, $h_{i}\left(t\right)$, and $M_{ei}\left(t\right)$ reduced to the shafts of electric motors.

It is assumed that only the angle at the reducer output shaft $x_{1}\left(t\right)$ and the current of the electric motor rotor circuit $x_{3}\left(t\right)$ can be measured by the respective sensors. It can be seen from (Equation 2) that the electric servo actuators of the manipulator are described by third order nonlinear differential equations with substantially variable parameters $H_{i}\left(t\right)$, $h_{i}\left(t\right)$, and $M_{ei}\left(t\right)$. Some of these parameters can even change sign in certain regimes.

Note that $H_{i}\left(t\right)$, $h_{i}\left(t\right)$, and $M_{ei}\left(t\right)$ depend on the manipulator generalized coordinates, which are functions of time. The exact form of these functions is not essential for SMO design; for simulation, we use the exact expression (see example). For simplicity, we use the notations $H_{i}\left(t\right)$, $h_{i}\left(t\right)$, and $M_{ei}\left(t\right)$.

It is assumed that the function $d\left(t\right)=-\frac{\tilde{M}\left(t\right)}{J_{H}+H^{*}\left(t\right)}$ corresponds to the unknown torque $\tilde{M}\left(t\right)$ due to increase in the Coulomb or viscous friction. The problem is to design a sliding mode observer estimating the function d(t).

The main contribution of this paper is that SMOs are constructed for robot manipulators described by nonstationary nonlinear dynamic models not satisfying matching, minimum phase, and detectability conditions. This is a result of the fact that SMO is not constructed for the original system but for its reduced order model invariant with respect to the disturbance. As a result, such a model may be free of some special peculiarities of the original system, which may prevent the possibility of designing SMO, in particular, the original system may be nonminimum phase or non-detectable, while its reduced order model will be minimum phase and detectable. In addition, the disturbance does not affect the observer, and its dimension becomes less than that of the original system. In addition, the linear part of the observer is designed based on the canonical model with constant parameters. This enables overcoming the difficulties related to non-stationarity and avoiding the complicated analysis, which is used in known papers to proof a convergence. All these advantages enable extension of a class of systems for which a sliding mode observer can be designed. As a result, this offers the practical possibility of solving the fault identification problems for those practical devices, which were a priori impossible for previous methods.

The paper is relevant to the Special Issue “Smart Sensor-Based Robot Control and Calibration” since sensor-based control of different robotics systems is one of the main challenges of modern robotics, and fault tolerant control can be achieved by fault identification.

The rest of the paper is organized as follows. In Section 2, the basic models are considered. In Section 3, the reduced model of the initial system is designed. The reduced order model transformation is considered in Section 4. Section 5 describes the SMO design. A practical example is considered in Section 6. Section 7 concludes the paper.

## 2. Preliminaries

In order to develop the new method of SMO design, we consider a system described by a general nonlinear nonstationary dynamic model under faults and disturbances:(3)$$\begin{array}{ccc} \dot{x}\left(t\right) & = & F(t)x(t)+G(t)u(t)+C(t)\Psi (x(t),u(t))+Dd(t)+L\rho (t),\\ y(t) & = & Hx(t), \end{array}$$
where $x\left(t\right)\in R^{n}$, $u\left(t\right)\in R^{m}$, $y\left(t\right)\in R^{l}$ are vectors of state, control, and output; F(t), G(t), and C(t) are known time varying matrices; *H*, *D*, and *L* are known constant matrices; d(t)∈R is a function describing faults: if there are no faults, d(t)=0, if a fault occurs, d(t) becomes an unknown function of time; $\rho \left(t\right)\in R^{p}$ is the unmatched disturbance, it is assumed that ρ(t) is an unknown bounded function of time; and Ψ(x,u) is the nonlinear term:(4)$$\Psi \left(x,u\right)=\left(\begin{array}{c} \varphi_{1}\left(A_{1}x,u\right)\\ \ldots \\ \varphi_{q}\left(A_{q}x,u\right) \end{array}\right),$$
$A_{1}$, …, $A_{q}$ are constant matrices, and $\varphi_{1},$ …, $\varphi_{q}$ are nonlinear functions. It is assumed that the function Ψ(x,u) satisfies the generalized Lipschitz condition about *x* uniformly for *t* and *u*:(5)$$∥\Psi \left(x,u\right)-\Psi \left(x^{\prime },u\right)∥\leq N∥x-x^{\prime }∥+M,$$
where N,M>0 are some constants. This assumption is typical for papers devoted to the fault identification problem via SMO.

It is assumed in [12] and many other papers that system (Equation 3) satisfies the following conditions: (1) matching condition when rank(H[LD])=rank([LD]), and (2) minimum phase condition when all invariant zeros of (F,[LD],H) lie in the left half plane. In [25,26], the system should be detectable. To solve the problem of fault identification, these conditions are not used in the present paper. The foundation of the suggested approach is the reduced order model of the original system; such a model can be free from some of the specific properties of the original system preventing SMO design, for example, the original system can be non-detectable while its reduced order model is detectable.

Note that the assumption d(t)∈R means that our approach can be applied to solve the problem of single fault identification as the most probable faults in the system. On the other hand, this assumption enables reduction of the limitations imposed on the original system in comparison with the abovementioned papers.

## 3. Reduced Order Model Design

Assuming that $x_{*}\in R^{k}$, k<n, is the state vector of the reduced order model, we set
(6)$$x_{*}\left(t\right)=\Phi \left(t\right)x\left(t\right)$$
for some differentiable matrix function Φ(t).

**Assumption** **1.**
*The function $\dot{\Phi }\left(t\right)x\left(t\right)$ is expressed in terms of $x_{*}$ and y; that is,*

(7)
$$\dot{\Phi }\left(t\right)x\left(t\right)=\alpha \left(x_{*}\left(t\right),y\left(t\right),t\right)$$

*for some function α.*


Solution of the problem is based on the reduced order model of system (Equation 3), generally described by the equations
(8)$$\begin{array}{ccc} {\dot{x}}_{*}\left(t\right) & = & F_{*}x_{*}\left(t\right)+G_{*}\left(t\right)u\left(t\right)+J_{*}\left(t\right)y\left(t\right)+C_{*}\left(t\right)\Psi \left(x_{*}\left(t\right),y\left(t\right),u\left(t\right)\right)\\ & \; & +\alpha \left(x_{*}\left(t\right),y\left(t\right),t\right)+D_{*}\left(t\right)d\left(t\right)+L_{*}\left(t\right)\rho \left(t\right),\\ y_{*}\left(t\right) & = & H_{*}x_{*}\left(t\right), \end{array}$$
where $x_{*}\left(t\right)\in R^{k}$ is the state vector, $F_{*}$, $G_{*}\left(t\right)$, $J_{*}\left(t\right)$, $C_{*}\left(t\right)$, $H_{*}$, $D_{*}\left(t\right)$, and $L_{*}\left(t\right)$ are matrices and matrix functions to be determined;
(9)$$C_{*}\left(t\right)\Psi \left(x_{*},y,u\right)=\left(\begin{array}{c} \varphi_{i_{1}}\left(A_{*1i_{1}}\left(t\right)x_{*}+A_{*2i_{1}}\left(t\right)y,u\right)\\ \ldots \\ \varphi_{i_{k}}\left(A_{*1i_{k}}\left(t\right)x_{*}+A_{*2i_{k}}\left(t\right)y,u\right) \end{array}\right),$$
and $A_{*1i_{1}}\left(t\right)$, $A_{*2i_{1}}\left(t\right)$, …, $A_{*1i_{k}}\left(t\right)$, $A_{*2i_{k}}\left(t\right)$ are matrix functions to be determined.

By analogy with (Equation 6), it is assumed that $y_{*}\left(t\right)=R_{*}y\left(t\right)$ for some matrix $R_{*}$. It is known [35,36] that matrices $R_{*}$ and Φ(t) satisfy the conditions
(10)$$\begin{array}{ccc} \Phi (t)F(t) & = & F_{*}\Phi \left(t\right)+J_{*}\left(t\right)H,\;R_{*}H=H_{*}\Phi \left(t\right),\;\Phi \left(t\right)G\left(t\right)=G_{*}\left(t\right),\\ \Phi (t)D & = & D_{*}\left(t\right),\;\Phi \left(t\right)L=L_{*}\left(t\right),\;\Phi \left(t\right)C=C_{*}\left(t\right)\\ A_{i} & = & \left(A_{*1i}\left(t\right)\;\;A_{*2i}\left(t\right)\right)\left(\begin{array}{c} \Phi (t)\\ H \end{array}\right),\;i=i_{1},\ldots ,i_{k}. \end{array}$$

Consider the method to solve these conditions and to construct the model (Equation 8) invariant with respect to the disturbance that enables solving the problem of exact fault identification. Note that if such a model does not exist, the problem of approximate fault identification can be solved [13].

The matrices $F_{*}$ and $H_{*}$ are sought in the canonical form
(11)$$F_{*}=\left(\begin{array}{ccccc} 0 & 1 & 0 & \ldots & 0\\ 0 & 0 & 1 & \ldots & 0\\ \ldots & \ldots & \ldots & \ldots & \\ 0 & 0 & 0 & \ldots & 0 \end{array}\right),\;H_{*}=\left(\begin{array}{ccccc} 1 & 0 & 0 & \ldots & 0 \end{array}\right).$$

Clearly, this is always possible if $\left(F_{*},H_{*}\right)$ is observable. If $\left(F_{*},H_{*}\right)$ is unobservable, system (Equation 8) can be transformed into observable canonical form [37], and then the matrices describing the observable part of this form can be presented in the canonical form (Equation 11) of less dimension.

Using these matrices, one obtains from (Equation 10) equations for rows of the matrices Φ(t) and $J_{*}\left(t\right)$:(12)$$\begin{array}{ccc} \Phi_{1} & = & R_{*}H,\\ \Phi_{i}\left(t\right)F\left(t\right) & = & \Phi_{i+1}\left(t\right)+J_{*i}\left(t\right)H,\;i=1,\ldots ,k-1,\\ \Phi_{k}\left(t\right)F\left(t\right) & = & J_{*k}\left(t\right)H, \end{array}$$
where $\Phi_{i}\left(t\right)$ and $J_{*i}\left(t\right)$ are *i*-th rows of the matrices Φ(t) and $J_{*}\left(t\right)$, i=1,…,k. As is shown in [35], Equations (Equation 12) can be transformed into the single equation
(13)$$\left(\begin{array}{cccc} R_{*} & -J_{*1}\left(t\right) & \ldots & -J_{*k}\left(t\right) \end{array}\right)W^{\left(k\right)}\left(t\right)=0,$$
where
(14)$$W^{\left(k\right)}\left(t\right)=\left(\begin{array}{c} HF^{k}\left(t\right)\\ HF^{k-1}\left(t\right)\\ \ldots \\ H \end{array}\right).$$

The condition Φ(t)L=0 of invariance with respect to the disturbance can be taken into account in the form $\left(\begin{array}{cccc} R_{*} & -J_{*1}\left(t\right) & \ldots & -J_{*k}\left(t\right) \end{array}\right)L^{\left(k\right)}\left(t\right)=0$ [35,36] where
(15)$$L^{\left(k\right)}\left(t\right)=\left(\begin{array}{cccc} HL & HF(t)L & \ldots & HF^{k-1}\left(t\right)L\\ 0 & HL & \ldots & HF^{k-2}\left(t\right)L\\ \ldots & \ldots & \ldots & \ldots \\ 0 & 0 & \ldots & 0 \end{array}\right).$$

The last equation and (Equation 13) result in the single equation
(16)$$\left(\begin{array}{cccc} R_{*} & -J_{*1}\left(t\right) & \ldots & -J_{*k}\left(t\right) \end{array}\right)\left(W^{\left(k\right)}\left(t\right)\;\;L^{\left(k\right)}\left(t\right)\right)=0.$$

Equation (Equation 16) has a nontrivial solution if
(17)$$rank\left(W^{\left(k\right)}\;\;L^{\left(k\right)}\right)<l\left(k+1\right).$$

To construct the model, find from (Equation 17) the minimal dimension *k*, and find the row $\left(\begin{array}{cccc} R_{*} & -J_{*1}\left(t\right) & \ldots & -J_{*k}\left(t\right) \end{array}\right)$ satisfying (Equation 16). Then calculate the rows of the matrix Φ(t) based on (Equation 12), and check the condition (Equation 4) for some function α. If it is true, calculate the matrix (Equation 9), and check the condition
(18)$$rank\left(\begin{array}{c} \Phi (t)\\ H \end{array}\right)=rank\left(\begin{array}{c} \Phi (t)\\ H\\ A_{i} \end{array}\right),\;\;i=i_{1},\ldots ,i_{k}.$$

If it is true, set $G_{*}\left(t\right):=\Phi \left(t\right)G$ and $D_{*}\left(t\right):=\Phi \left(t\right)D$; the matrices $A_{*1i}\left(t\right)$ and $A_{*2i}\left(t\right)$, $i=i_{1},\ldots ,i_{k}$, are found from (Equation 10). If (Equation 18) is not true, one finds another solution of (Equation 16) with former or incremented dimension *k*. If (Equation 18) is not true for all k<n, the model invariant with respect to the disturbance cannot be designed.

## 4. Reduced Order Model Transformation

Write down all matrices in (Equation 8) in the form
(19)$$\begin{array}{ccc} F_{*} & = & \left(\begin{array}{cc} F_{1} & F_{2}\\ F_{3} & F_{4} \end{array}\right),\;H_{*}=\left(1\;\;\;\;\;0\right),\;G_{*}\left(t\right)=\left(\begin{array}{c} G_{*1}\left(t\right)\\ G_{*2}\left(t\right) \end{array}\right),\\ J_{*}\left(t\right) & = & \left(\begin{array}{c} J_{*1}\left(t\right)\\ J_{*2}\left(t\right) \end{array}\right),\;C_{*}\left(t\right)=\left(\begin{array}{c} C_{*1}\left(t\right)\\ C_{*2}\left(t\right) \end{array}\right),\;D_{*}\left(t\right)=\left(\begin{array}{c} D_{*1}\left(t\right)\\ D_{*2}\left(t\right) \end{array}\right),\\ \Phi (t) & = & \left(\begin{array}{c} \Phi^{\left(1\right)}\left(t\right)\\ \Phi^{\left(2\right)}\left(t\right) \end{array}\right)=\left(\begin{array}{c} R_{*}H\\ \Phi^{\left(2\right)}\left(t\right) \end{array}\right),\;\alpha =\left(\begin{array}{c} \alpha_{1}\\ \alpha_{2} \end{array}\right)=\left(\begin{array}{c} 0\\ \alpha_{2} \end{array}\right), \end{array}$$ where (20)$$\begin{array}{ccc} F_{1} & = & 0,\;F_{2}=\left(1\;\;\;\;0\;\;\;\;\ldots \;\;\;\;0\;\;\;0\right)\in R^{1\times k-1},\\ F_{3} & = & 0,\;F_{4}=\left(\begin{array}{ccccc} 0 & 1 & 0 & \ldots & 0\\ 0 & 0 & 1 & \ldots & 0\\ \ldots & \ldots & \ldots & \ldots & \ldots \\ 0 & 0 & 0 & \ldots & 0 \end{array}\right)\in R^{k-1\times k-1}, \end{array}$$
the rest of the matrices in (Equation 19) have the appropriate dimensions. The function $\alpha_{1}$ is equal to zero since
(21)$$\alpha_{1}\left(x_{*}\left(t\right),y\left(t\right),t\right)={\dot{\Phi }}_{1}\left(t\right)x\left(t\right)=\frac{d(R_{*}H)}{dx}x\left(t\right)=0.$$

Introduce a coordinate transformation $z=Tx_{*}$ with $T=\left(\begin{array}{cc} 1 & 0\\ Q & I_{k-1} \end{array}\right),$ where $Q\in R^{k-1\times 1}$ is selected to make ${\bar{F}}_{4}=F_{4}+QF_{2}$ stable. Since $\left(F_{4},F_{2}\right)$ is observable, this matrix exists and is of the form $Q:={\left(a_{1}\;\;a_{2}\;\;\ldots \;\;a_{k-1}\right)}^{T}$. As a result, the model (Equation 8) takes the form
(22)$$\begin{array}{ccc} {\dot{z}}_{1}\left(t\right) & = & {\bar{F}}_{1}y_{*}\left(t\right)+{\bar{F}}_{2}z_{2}\left(t\right)+{\bar{G}}_{1}\left(t\right)u\left(t\right)+{\bar{C}}_{1}\left(t\right)\Psi \left(z_{2}\left(t\right),y_{*}\left(t\right),y\left(t\right),u\left(t\right)\right)\\ & \; & +{\bar{J}}_{1}\left(t\right)y\left(t\right)+{\bar{D}}_{1}\left(t\right)d\left(t\right),\\ {\dot{z}}_{2}\left(t\right) & = & {\bar{F}}_{3}y_{*}\left(t\right)+{\bar{F}}_{4}z_{2}\left(t\right)+{\bar{G}}_{2}\left(t\right)u\left(t\right)+{\bar{C}}_{2}\left(t\right)\Psi \left(z_{2}\left(t\right),y_{*}\left(t\right),y\left(t\right),u\left(t\right)\right)\\ & \; & +{\bar{J}}_{2}\left(t\right)y\left(t\right)+{\bar{D}}_{2}\left(t\right)d\left(t\right)+{\bar{\alpha }}_{2}\left(z_{2}\left(t\right),y_{*}\left(t\right),y\left(t\right),u\left(t\right),t\right),\\ y_{*}\left(t\right) & = & z_{1}\left(t\right), \end{array}$$
where ${\bar{F}}_{1}=-a_{1},\;{\bar{F}}_{2}=\left(1\;\;0\;\;\ldots \;\;0\;\;0\right),$
(23)$$\begin{array}{ccc} {\bar{F}}_{3} & = & -\left(\begin{array}{c} a_{1}^{2}+a_{2}\\ a_{1}a_{2}+a_{3}\\ \ldots \\ a_{1}a_{k-1} \end{array}\right),\;{\bar{F}}_{4}=\left(\begin{array}{ccccc} a_{1} & 1 & 0 & \ldots & 0\\ a_{2} & 0 & 1 & \ldots & 0\\ \ldots & \ldots & \ldots & \ldots & \ldots \\ a_{k-1} & 0 & 0 & \ldots & 0 \end{array}\right),\\ {\bar{G}}_{1}\left(t\right) & = & G_{*1}\left(t\right),\;{\bar{G}}_{2}\left(t\right)=QG_{*1}\left(t\right)+G_{*2}\left(t\right),\\ {\bar{J}}_{1}\left(t\right) & = & J_{*1}\left(t\right),\;{\bar{J}}_{2}\left(t\right)=QJ_{*1}\left(t\right)+J_{*2}\left(t\right),\\ {\bar{C}}_{1}\left(t\right) & = & C_{*1}\left(t\right),\;{\bar{C}}_{2}\left(t\right)=QC_{*1}\left(t\right)+C_{*2}\left(t\right),\\ {\bar{D}}_{1}\left(t\right) & = & D_{*1}\left(t\right),\;{\bar{D}}_{2}\left(t\right)=QD_{*1}\left(t\right)+D_{*2}\left(t\right),\\ \; & \; & {\bar{\alpha }}_{2}\left(z_{2},y_{*},y,u,t\right)=\alpha_{2}\left(z_{2},y_{*},y,u,t\right). \end{array}$$

Note that the model (Equation 22) corresponds to that in [8,26] and other similar papers where the matrix $F_{*2}$ is stable due to the minimum phase or detectability properties of the original system; in our approach the matrix $F_{*2}$ is stable because of the canonical form of the matrices $F_{*}$ and $H_{*}$.

## 5. Sliding Mode Observer Design

Since ${\bar{F}}_{4}$ is stable, symmetric positive definite matrices *P* and *W* exist such that ${\bar{F}}_{4}^{T}P+P{\bar{F}}_{4}=-W.$ By analogy with [26], SMO is sought in the form
(24)$$\begin{array}{ccc} {\dot{\hat{z}}}_{1} & = & {\bar{F}}_{1}y_{*}+{\bar{F}}_{2}{\hat{z}}_{2}+{\bar{G}}_{1}u+{\bar{J}}_{1}y+{\bar{C}}_{1}\Psi \left({\hat{z}}_{2},y_{*},y,u\right)+k_{2}e_{1}+k_{3}v,\\ {\dot{\hat{z}}}_{2} & = & {\bar{F}}_{3}y_{*}+{\bar{F}}_{4}{\hat{z}}_{2}+{\bar{G}}_{2}u+{\bar{J}}_{2}y+{\bar{C}}_{2}\Psi \left({\hat{z}}_{2},y_{*},y,u\right)+{\bar{\alpha }}_{2}\left({\hat{z}}_{2},y_{*},y,u,t\right)+{\bar{K}}_{1}v,\\ {\hat{y}}_{*} & = & {\hat{z}}_{1}, \end{array}$$
where $v=sign(e_{1})$, $e_{1}=y_{*}-{\hat{y}}_{*}$, ${\bar{K}}_{1}=P^{-1}{\bar{F}}_{2}^{T}k_{1}$, $k_{1},k_{2}$, and $k_{3}$ are positive numbers.

Let $e_{2}=z_{2}-{\hat{z}}_{2}$; it follows from (Equation 22) and (Equation 24)
(25)$$\begin{array}{ccc} {\dot{e}}_{1} & = & {\bar{F}}_{2}e_{2}+{\bar{C}}_{1}\Delta \Psi +{\bar{D}}_{1}d-k_{2}e_{1}-k_{3}v,\\ {\dot{e}}_{2} & = & {\bar{F}}_{4}e_{2}+{\bar{C}}_{2}\Delta \Psi +\Delta \alpha +{\bar{D}}_{2}d-{\bar{K}}_{1}v, \end{array}$$
where $\Delta \Psi =\Psi \left(z_{2},y_{*},y,u\right)-\Psi \left({\hat{z}}_{2},y_{*},y,u\right),$ and $\;\Delta \alpha ={\bar{\alpha }}_{2}\left(z_{2},y_{*},y,u,t\right)-{\bar{\alpha }}_{2}\left({\hat{z}}_{2},y_{*},y,u,t\right).$ Assume that the functions Ψ(x,u) and $\alpha (x_{*},y,t)$ satisfy the Lipschitz condition (Equation 5) about *x*, and $x_{*}$, respectively, then the functions $\Psi (z_{2},y_{*},y,u)$ and ${\bar{\alpha }}_{2}\left(z_{2},y_{*},y,u,t\right)$ satisfy this condition and
(26)$$\begin{array}{ccc} ∥{\bar{C}}_{1}\Delta \Psi ∥ & \leq & N_{*1}∥e_{2}∥+M_{*1},\\ ∥{\bar{C}}_{2}\Delta \Psi +\Delta \alpha ∥ & \leq & N_{*2}∥e_{2}∥+M_{*2} \end{array}$$
for some positive $N_{*1}$, $N_{*2}$, $M_{*1},$ and $M_{*2}$.

**Theorem** **1.**
*Assume that $\underline{\lambda }\left(W\right)\geq 2∥P∥N_{*2}.$ If ${\bar{D}}_{1}=0$, the function d(t) can be estimated by*

(27)
$$\hat{d}\left(t\right)={\bar{D}}_{2}^{+}{\bar{K}}_{1}v_{eq}\left(t\right),$$

*if ${\bar{D}}_{1}\neq 0$, d(t) can be estimated by*

(28)
$$\hat{d}\left(t\right)=k_{3}{\bar{D}}_{1}^{+}v_{eq}\left(t\right),$$

*where ${\bar{D}}_{1}^{+}={\left({\bar{D}}_{1}^{T}{\bar{D}}_{1}\right)}^{-1}{\bar{D}}_{1}^{T}$ and ${\bar{D}}_{2}^{+}={\left({\bar{D}}_{2}^{T}{\bar{D}}_{2}\right)}^{-1}{\bar{D}}_{2}^{T},$ $v_{eq}\left(t\right)$ is the so-called equivalent output injection signal representing the average behavior of the discontinuous function v(t). Similar to [8], we use as $v_{eq}\left(t\right)$ the continuous approximation*

(29)
$$v_{eq}\left(t\right)=\frac{e_{1}\left(t\right)}{|e_{1}\left(t\right)|+\varepsilon },$$

*where ε is a small positive scalar.*


**Proof** **of** **Theorem 1.**By analogy with [26], we prove firstly that $∥e_{2}∥\leq \delta =max\left\{\delta_{1},\delta_{2}\right\}$, where
(30)$$\delta_{1}=\frac{2\bar{\lambda }\left(P\right)(\beta ∥P{\bar{D}}_{2}∥+∥P{\bar{K}}_{1}∥)}{\underline{\lambda }\left(P\right)(\underline{\lambda }\left(W\right)-2∥P∥N_{*2})},\;\delta_{2}=\sqrt{\frac{\bar{\lambda }\left(P\right)}{\underline{\lambda }\left(P\right)}}∥e_{2}\left(0\right)∥,$$
β is such that β≥∥d(t)∥. Consider the Lyapunov function $V_{2}=e_{2}^{T}Pe_{2}$, and find its derivative with respect to time, taking into account (Equation 25) and (Equation 26):
(31)$$\begin{array}{ccc} \dot{V_{2}} & = & -e_{2}^{T}We_{2}+2e_{2}^{T}P{\bar{D}}_{2}d-2e_{2}^{T}P{\bar{K}}_{1}v+2e_{2}^{T}P\left({\bar{C}}_{2}\Delta \Psi +\Delta \alpha \right)\\ & \leq & -∥e_{2}{∥}^{2}(\underline{\lambda }\left(W\right)-2∥P∥N_{*2})+2∥e_{2}∥(\beta ∥P{\bar{D}}_{2}∥+∥P{\bar{K}}_{1}∥+M_{*2}). \end{array}$$Using Rayleigh’s inequality $\underline{\lambda }\left(P\right)∥e_{2}{∥}^{2}\leq V_{2}\leq \bar{\lambda }\left(P\right){∥e_{2}∥}^{2},$ one obtains
(32)$$\begin{array}{ccc} {\dot{V}}_{2} & \leq & \frac{\underline{\lambda }\left(W\right)-2∥P∥N_{*2}}{\bar{\lambda }\left(P\right)}V_{2}+\frac{2\sqrt{V_{2}}}{\sqrt{\underline{\lambda }\left(P\right)}}(\beta ∥P{\bar{D}}_{2}∥+∥P{\bar{K}}_{1}∥+M_{*2}). \end{array}$$The rest of the proof coincides with that in [26].Secondly, we prove that by suitable choices of observer gains, $e_{1}=0$ in finite time and sliding motion is achieved. Consider the Lyapunov function $V_{1}=e_{1}^{1}$, and find its derivative with respect to time taking into account (Equation 25):
(33)$$\dot{V_{1}}=2e_{1}{\dot{e}}_{1}=2e_{1}\left({\bar{F}}_{2}e_{2}+{\bar{C}}_{1}\Delta \Psi +{\bar{D}}_{1}d-k_{2}e_{1}-k_{3}v\right).$$Since $v=sign(e_{1})$, and $∥{\bar{C}}_{1}\Delta \Psi ∥\leq \delta N_{*1}+M_{*1}$, then $2e_{1}k_{3}v=2k_{3}\left|e_{1}\right|$, and
(34)$$\begin{array}{ccc} \dot{V_{1}} & \leq & -2k_{2}e_{1}^{2}+2|e_{1}|(-k_{3}+\delta N_{*1}+M_{*1}+∥{\bar{F}}_{2}∥∥e_{2}∥+∥{\bar{D}}_{1}∥∥d∥)\\ & \leq & -2k_{2}e_{1}^{2}+2|e_{1}|(-k_{3}+\delta N_{*1}+M_{*1}+\delta +\beta ∥{\bar{D}}_{1}∥). \end{array}$$If $k_{3}$ satisfies
(35)$$k_{3}\geq \beta ∥{\bar{D}}_{1}∥+\delta \left(N_{*1}+1\right)+M_{*1},$$
then it can be shown by analogy with [26] that $\dot{V_{1}}\leq -c_{1}\sqrt{V_{1}}$ for some $c_{1}>0$, and sliding motion ($e_{1}={\dot{e}}_{1}=0$) happens in finite time.Thirdly, to prove that by suitable choices the observer gains $e_{2}=0$ in finite time and sliding motion is achieved, consider the Lyapunov function $V_{2}$ and its derivative (Equation 31). From the first equation in (Equation 25), and since sliding motion has occurred ($e_{1}={\dot{e}}_{1}=0$), it follows that ${\bar{F}}_{2}e_{2}=k_{3}v-{\bar{D}}_{1}d-{\bar{C}}_{1}\Delta \Psi .$ Using ${\bar{K}}_{1}=P^{-1}{\bar{F}}_{2}^{T}k_{1}$, we obtain
(36)$$\begin{array}{ccc} \dot{V_{2}} & = & -e_{2}^{T}We_{2}+2e_{2}^{T}P\left({\bar{C}}_{2}\Delta \Psi +\Delta \alpha \right)+2e_{2}^{T}P{\bar{D}}_{2}d-2e_{2}^{T}{\bar{F}}_{2}^{T}k_{1}v\\ & = & -e_{2}^{T}We_{2}+2\left(e_{2}^{T}P\left({\bar{C}}_{2}\Delta \Psi +\Delta \alpha \right)+e_{2}^{T}P{\bar{D}}_{2}d-{\left(k_{3}v-{\bar{D}}_{1}d-{\bar{C}}_{1}\Delta \Psi \right)}^{T}k_{1}v\right). \end{array}$$Since $∥e_{2}\left(t\right)∥\leq \delta$, it follows that
(37)$$\begin{array}{c} \dot{V_{2}}\leq -e_{2}^{T}We_{2}+2(\delta ∥P∥\left(\delta N_{*2}+M_{*2}\right)+\beta \delta ∥P{\bar{D}}_{2}∥-k_{1}k_{3}+k_{1}\beta ∥{\bar{D}}_{1}∥+k_{1}\left(\delta N_{*1}+M_{*1}\right)). \end{array}$$If $k_{3}$ and $k_{1}$ are chosen, respectively, as
(38)$$\begin{array}{ccc} k_{3} & > & \beta ∥{\bar{D}}_{1}∥+\delta N_{*1}+M_{*1},\\ k_{1} & > & \frac{\delta (∥P∥\left(\delta N_{*2}+M_{*2}\right)+\beta ∥P{\bar{D}}_{2}∥)}{k_{3}-\beta ∥{\bar{D}}_{1}∥-\delta N_{*1}-M_{*1}}, \end{array}$$
then it can be shown by analogy with [26] that $\dot{V_{2}}\leq -c_{2}\sqrt{V_{2}}$ for some $c_{2}>0$, and finite convergence of $e_{2}$ happens as well. Based on (Equation 35) and (Equation 38), one has to choose $k_{3}$ as
(39)$$\begin{array}{c} k_{3}>\beta ∥{\bar{D}}_{1}∥+\delta \left(N_{*1}+1\right)+M_{*1}. \end{array}$$It follows from (Equation 25) that if ${\bar{D}}_{1}=0$, then the function d(t) can be estimated from the second equation in (Equation 25) as (Equation 27); otherwise, we use the first equations in (Equation 25) and obtain (Equation 28). Theorem has been proved. □

## 6. Practical Example

Consider the robot manipulator PUMA presented in Figure 1 and its actuator (Equation 2) described by the following matrices: (40)$$\begin{array}{ccc} F & = & \left(\begin{array}{ccc} 0 & \frac{1}{i_{r}} & 0\\ 0 & -\frac{K_{v}+h^{*}\left(t\right)}{J_{H}+H^{*}\left(t\right)} & \frac{K_{m}}{J_{H}+H^{*}\left(t\right)}\\ 0 & -\frac{K_{\omega }}{L_{m}} & -\frac{R_{m}}{L_{m}} \end{array}\right),\;G=\left(\begin{array}{c} 0\\ 0\\ \frac{K_{u}}{L_{m}} \end{array}\right),\;D=C=\left(\begin{array}{c} 0\\ 1\\ 0 \end{array}\right),\\ H & = & \left(\begin{array}{ccc} 1 & 0 & 0\\ 0 & 0 & 1 \end{array}\right),\;\;A=\left(0\;\;1\;\;0\right),\;\varphi \left(x,u\right)=-\frac{M_{e}^{*}}{J_{H}+H^{*}\left(t\right)}-\frac{M_{f}}{J_{H}+H^{*}\left(t\right)}sign\left(Ax\left(t\right)\right). \end{array}$$

The functions $H^{*}\left(t\right)$, $h^{*}\left(t\right)$, and $M_{e}^{*}\left(t\right)$ are described as follows: (41)$$\begin{array}{ccc} H^{*} & = & 1.6324+0.9cos\left(q_{3}\right),\;h^{*}=-0.9{\dot{q}}_{3}sin\left(q_{3}\right),\\ M_{e}^{*} & = & \left(0.75+0.45cos\left(q_{3}\right)\right){\ddot{q}}_{3}-0.45{\dot{q}}_{3}^{2}sin\left(q_{3}\right)+26.166sin\left(q_{2}\right)+8.82sin\left(q_{2}+q_{3}\right)\\ & \; & -0.5\left(1.291sin\left(2q_{2}\right)+0.333sin\left(2q_{2}+2q_{3}\right)+0.9sin\left(2q_{2}+q_{3}\right)\right){\dot{q}}_{1}^{2}. \end{array}$$

Clearly, rank(HD)=0≠rank(D)=1; therefore, the matching condition is not satisfied. Thus, the method suggested in [8,11] cannot be used in our case.

Construct a sliding mode observer estimating the function d(t) corresponding to the matrix *D*. The solution of (Equation 16) with L=0 is as follows: (42)$$\begin{array}{ccc} R_{*} & = & \left(0\;\;\;1\right),\;\;\Phi \left(t\right)=\left(\begin{array}{ccc} 0 & 0 & 1\\ 0 & -\frac{K_{\omega }}{L_{m}} & \frac{K_{v}+h^{*}\left(t\right)}{J_{H}+H^{*}\left(t\right)} \end{array}\right),\;\;G_{*}=\left(\begin{array}{c} \frac{K_{u}}{L_{m}}\\ \frac{K_{u}\left(K_{v}+h^{*}\left(t\right)\right)}{L_{m}\left(J_{H}+H^{*}\left(t\right)\right)} \end{array}\right),\\ J_{*} & = & \left(\begin{array}{cc} 0 & -\frac{\left(K_{v}+h^{*}\left(t\right)\right)L_{m}+\left(J_{H}+H^{*}\left(t\right)\right)R_{m}}{\left(J_{H}+H^{*}\left(t\right)\right)L_{m}}\\ 0 & -\frac{K_{m}K_{\omega }+\left(K_{v}+h^{*}\left(t\right)\right)R_{m}}{\left(J_{H}+H^{*}\left(t\right)\right)L_{m}} \end{array}\right),\;\;C_{*}=\left(\begin{array}{c} 0\\ -\frac{K_{\omega }}{L_{m}} \end{array}\right),\;\;D_{*}=\left(\begin{array}{c} 0\\ -\frac{K_{\omega }}{L_{m}} \end{array}\right). \end{array}$$

It is assumed that the function $d\left(t\right)=-\frac{\tilde{M}\left(t\right)}{J_{H}+H^{*}\left(t\right)}$ corresponds to the unknown torque $\tilde{M}\left(t\right)$ due to increase in the Coulomb or viscous friction.

Clearly, the function α depends only on $y_{2}$, and since ${\dot{H}}^{*}\left(t\right)=h^{*}\left(t\right)$, then
(43)$$\alpha \left(y\left(t\right),t\right)=\left(\begin{array}{c} 0\\ \frac{{\dot{h}}^{*}\left(t\right)\left(J_{H}+H^{*}\left(t\right)\right)-h^{*}\left(t\right)\left(K_{v}+h^{*}\left(t\right)\right)}{{\left(J_{H}+H^{*}\left(t\right)\right)}^{2}}y_{2}\left(t\right) \end{array}\right).$$

The model is described as follows:(44)$$\begin{array}{ccc} {\dot{z}}_{1} & = & z_{2}-\frac{\left(K_{v}+h^{*}\left(t\right)\right)L_{m}+\left(J_{H}+H^{*}\left(t\right)\right)R_{m}}{\left(J_{H}+H^{*}\left(t\right)\right)L_{m}}y_{2}\left(t\right)+\frac{K_{u}}{L_{m}}u\left(t\right),\\ {\dot{z}}_{2} & = & -\frac{K_{m}K_{\omega }+\left(K_{v}+h^{*}\left(t\right)\right)R_{m}}{\left(J_{H}+H^{*}\left(t\right)\right)L_{m}}y_{2}\left(t\right)-\frac{K_{\omega }}{L_{m}}d\left(t\right)+\alpha \left(y\left(t\right),t\right)+\frac{K_{u}\left(K_{v}+h^{*}\left(t\right)\right)}{L_{m}\left(J_{H}+H^{*}\left(t\right)\right)}u\left(t\right)\\ & \; & -\frac{K_{\omega }}{L_{m}}\left(-\frac{M_{e}^{*}}{J_{H}+H^{*}\left(t\right)}-\frac{M_{f}}{J_{H}+H^{*}\left(t\right)}sign\left(\left(-z_{2}\left(t\right)+\frac{K_{u}\left(K_{v}+h^{*}\left(t\right)\right)}{L_{m}\left(J_{H}+H^{*}\left(t\right)\right)}y_{2}\left(t\right)\right)\frac{L_{m}}{K_{\omega }}\right)\right),\\ y_{*} & = & z_{1}. \end{array}$$

Clearly, based on (Equation 26), we have for the function “sign” $N_{*}=0$ and $M_{*}=2$. Since $F_{2}=1$ and $F_{4}=0$, set Q=−1 and ${\bar{F}}_{4}=-1$, which gives P=1 and W=2. The description of SMO is given by
(45)$$\begin{array}{ccc} {\dot{\hat{z}}}_{1}\left(t\right) & = & {\hat{z}}_{1}\left(t\right)+{\hat{z}}_{2}\left(t\right)+\frac{K_{u}}{L_{m}}u\left(t\right)+k_{2}e_{2}\left(t\right)+k_{3}v\left(t\right)-\frac{\left(K_{v}+h^{*}\left(t\right)\right)L_{m}+\left(J_{H}+H^{*}\left(t\right)\right)R_{m}}{\left(J_{H}+H^{*}\left(t\right)\right)L_{m}}y_{2}\left(t\right),\\ {\dot{\hat{z}}}_{2}\left(t\right) & = & -\frac{K_{m}K_{\omega }+\left(K_{v}+h^{*}\left(t\right)\right)R_{m}}{\left(J_{H}+H^{*}\left(t\right)\right)L_{m}}y_{2}\left(t\right)+\alpha \left(y\left(t\right),t\right)+\frac{K_{u}\left(K_{v}+h^{*}\left(t\right)\right)}{L_{m}\left(J_{H}+H^{*}\left(t\right)\right)}u\left(t\right)\\ & \; & -\frac{K_{\omega }}{L_{m}}\left(-\frac{M_{e}^{*}}{J_{H}+H^{*}\left(t\right)}-\frac{M_{f}}{J_{H}+H^{*}\left(t\right)}sign\left(\left(-{\hat{z}}_{2}\left(t\right)+\frac{K_{u}\left(K_{v}+h^{*}\left(t\right)\right)}{L_{m}\left(J_{H}+H^{*}\left(t\right)\right)}y_{2}\left(t\right)\right)\frac{L_{m}}{K_{\omega }}\right)\right)\\ \;\; & \; & -{\hat{z}}_{1}\left(t\right)-{\hat{z}}_{2}\left(t\right)-\frac{K_{u}}{L_{m}}u\left(t\right)+{\bar{K}}_{1}v\left(t\right)+\frac{\left(K_{v}+h^{*}\left(t\right)\right)L_{m}-\left(J_{H}+H^{*}\left(t\right)\right)R_{m}}{\left(J_{H}+H^{*}\left(t\right)\right)L_{m}}y_{2}\left(t\right),\\ {\hat{y}}_{*}\left(t\right) & = & {\hat{z}}_{1}\left(t\right), \end{array}$$
where $v\left(t\right)=sign(y_{*}\left(t\right)-{\hat{y}}_{*}\left(t\right))$, ${\bar{K}}_{1}=P^{-1}{\bar{F}}_{2}^{T}k_{1}=k_{1}$. The function d(t) is estimated as $\hat{d}\left(t\right)=k_{1}D_{2}^{+}v_{eq}\left(t\right)$.

To check the effectiveness of the observer, consider the servo actuator in the second DOF of the manipulator PUMA (Figure 1). For simulation, the nominal values of the parameters are taken as follows: $i_{r}=100$, $J_{H}=0.0001$ kg·m^2^, $K_{\omega }=0.04$ V·s, $K_{u}=100$, R=0.5
Ω, L=0.0005 H, $K_{m}=0.04$ N·m/A, $K_{v}=0.005$ Nms/rad, $K_{f}=0.02$ Nm, and u(t)=2sin(t). The coefficients $k_{1},k_{2}$, and $k_{3}$ are $k_{1}=10^{7}$, $k_{2}=1$, and $k_{3}=10^{4}.$ The fault is modeled by increasing $K_{v}\left(t\right)$ and $M_{f}\left(t\right)$ by 50% since 3s in $\tilde{M}\left(t\right)=K_{v}\left(t\right)+M_{f}\left(t\right)$; the variables $q_{1}$, $q_{2}$, and $q_{3}$ are modeled as follows: $q_{1}=2sin\left(1.5t\right)$, $q_{2}=sin\left(t\right)$, and $q_{3}=sin\left(2t\right)$.

The simulation results are demonstrated in Figure 2 and Figure 3 showing the behavior of the functions M(t) and $\tilde{M}\left(t\right)$ and the estimation error $\Delta \tilde{M}\left(t\right)=\tilde{M}\left(t\right)-M\left(t\right)$, respectively. Clearly, the estimation error is rather small, which shows a high quality of estimation.

Note that the quantization error/measurement noise of the system can be taken into consideration by modification of the fault identification procedure. The limited space of the paper does not allow us to consider this in full measure; such a problem was considered in [38,39].

## 7. Conclusions

In this paper, the problem of fault identification in robot manipulators described by nonstationary nonlinear dynamic models under disturbances based on sliding mode observers has been studied. Distinguished from the known methods, the suggested approach was based on the reduced order model of the original system having different sensitivity to faults and disturbances. This model was realized in observable canonical form with constant parameters that enabled overcoming the difficulties related to non-stationarity and relaxing the limitation imposed on the original system. The theoretical results were illustrated by the practical example of the manipulator PUMA.

## Figures and Tables

**Figure 1 sensors-22-00317-f001:**
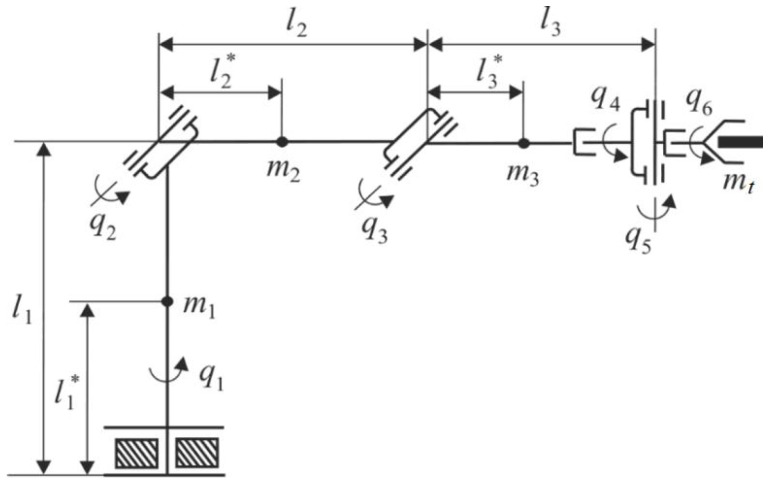
Kinematic scheme of manipulator: $m_{i}$ is the mass of *i*-th link; $l_{i}$ is the length of *i*-th link; $l_{i}^{*}$ is the distance from the rotation joint of *i*-th link to its center of mass; and $m_{t}$ is the mass of the load (tool in the gripper).

**Figure 2 sensors-22-00317-f002:**
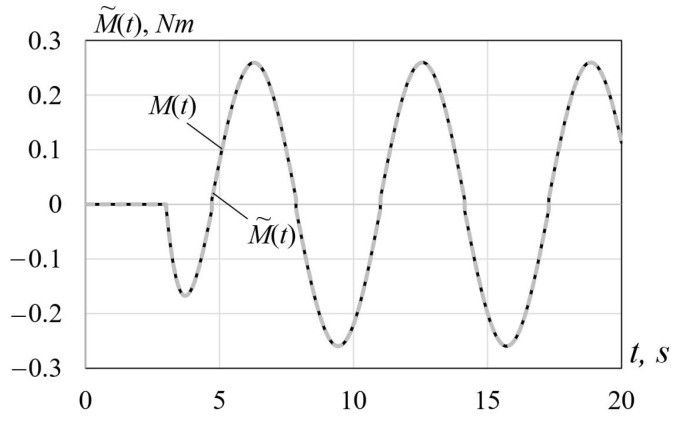
Behavior of the functions M(t) and $\tilde{M}\left(t\right)$.

**Figure 3 sensors-22-00317-f003:**
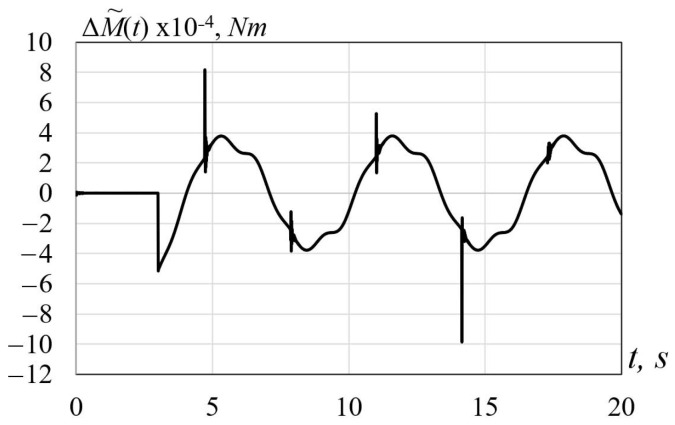
Behavior of the estimation error $\Delta \tilde{M}\left(t\right)=\tilde{M}\left(t\right)-M\left(t\right)$.

## Data Availability

There is no data set associated with the paper.

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
