# Peer review of "Fault Identification in Electric Servo Actuators of Robot Manipulators Described by Nonstationary Nonlinear Dynamic Models Using Sliding Mode Observers"

_sensors, 2022, doi:10.3390/s22010317_

Round 1
Reviewer 1 Report
Please refer to the attachment for details.

Author Response
Authors wish to thank the Reviewer for his comments, which helped us to improve our paper. All the comments of the reviewer were taken into account in the revised
version as detailed below.

Reviewer 2 Report
The manuscript titled ‘Fault Identification in Robot Manipulators Described by Non-stationary Nonlinear Dynamic Models Using Sliding Mode Observers’ introduces an approach based on the reduced model order of the original system or solving the problem of fault identification in electro servo actuators of robot manipulators. The use of a reduced model order gives the possibility to solve the fault identification problems for those practical devices which are impossible with previous methods, extending the class of systems for which sliding mode observer can be designed.
The methodology is theoretically shown, and it is applied to the model of commercial PUMA manipulator.
The results are interesting, the manuscript contains the mathematical demonstration of the proposed approach and it interesting for the scientific community; anyway the relevance of the manuscript with the journal and special issue topic should be clarified and demonstrated.
Author Response
Authors wish to thank the Reviewer for his kind response.

Reviewer 3 Report
In this paper, the problem of fault identification in robot manipulators described by non-stationary nonlinear dynamic models under disturbances based on sliding mode observers has been studied.
In general, authors present the fault identification in robots manipulators by nonlinear models using sliding mode observers. Authors should consider the following comments to clarify the main contributions of their paper.
1.- In the page 1, in the introduction, authors say “Sliding mode observers are used for unknown input estimation and fault identification (reconstruction) in different systems [8,13,18,24,25,28,32] and for fault tolerant control [1,14].”, they should include references [a]-[f] for the fault tolerant control, because the references consider the mentioned topic.
[a] Adapting H-Infinity Controller for the Desired Reference Tracking of the Sphere Position in the Maglev Process, Information Sciences, Vol. 569, 669-686, 2021.
[b] Transformed Structural Properties Method to Determine the Controllability and Observability of Robots, Applied Sciences, Vol. 11, No. 7, 3082, 2021.
[c] PI-Type Controllers and Σ–Δ Modulation for Saturated DC-DC Buck Power Converters, IEEE Access, Vol. 9, pp. 20346-20357, 2021.
[d] VSC-HVDC and its Applications for Black Start Restoration Processes, Applied Sciences-Basel, Vol. 11, No. 12, pp. 5648, 2021.
[e] PD Control Compensation Based on a Cascade Neural Network Applied to a Robot Manipulator, Frontiers in Neurorobotics, Vol. 14, 2020.
[f] Sensorless tracking control for a full-bridge Buck inverter-DC motor system: Passivity and flatness-based design, IEEE Access, Vol. 9, pp. 132191-132204, 2021.
2.- All the equations of the paper should have equation number.
3.- In the pages 4, 7, in the equation (9), (20), authors should clarify if there is a difficulty to solve these conditions.
4.- In the page 6, the definitions of F1, F2, F3, F4 should be better organized in order that they can fit in the page.
5.- In the page 7, in the equations (21), (22), authors should clarify that D1, D2 should be different to zero to avoid singularities in the estimation of the function d(t).
6.- In the page 8, in the practical example, authors should clarify if they compare their method with other previous.
Author Response

(The authors gave the same response as above.)

Round 2
Reviewer 1 Report
The authors addressed all of our concerns in the current form. However, one more reference (which is highly related) is suggested to include in the introduction section of the final form. The reference also addressed the external fault identification and reconstruction using the concept of equivalent control injection (in SMC) for a single link robot arm. External fault recovery validations were also illustrated in that paper.
Nonlinear integral type observer design for state estimation and unknown input reconstruction, by Peng, C. C., 2017, In: Applied Sciences (Switzerland). 7, 1, 67.
Author Response
Thanks for your comment. The reference has been included.
Reviewer 2 Report
Congratulatios for your work.
Author Response
We thank the Reviewer for his comment.